# Serial imaging of micro-agents and cancer cell spheroids in a microfluidic channel using multicolor fluorescence microscopy

**Mert Kaya**[1,2]*, **Fabian Stein**[3], **Jeroen Rouwkema**[3], **Islam S. M. Khalil**[2], **Sarthak Misra**[1,2]

**1** Surgical Robotics Laboratory, Department of Biomechanical Engineering, University of Twente, Enschede, The Netherlands, **2** Surgical Robotics Laboratory, Department of Biomedical Engineering and University Medical Centre Groningen, University of Groningen, Groningen, The Netherlands, **3** Vascularization Laboratory, Department of Biomechanical Engineering, University of Twente, Enschede, The Netherlands

* m.kaya@utwente.nl

**Data Availability Statement:** All relevant data are within the paper and its Supporting information files.

## Abstract

Multicolor fluorescence microscopy is a powerful technique to fully visualize many biological phenomena by acquiring images from different spectrum channels. This study expands the scope of multicolor fluorescence microscopy by serial imaging of polystyrene micro-beads as surrogates for drug carriers, cancer spheroids formed using HeLa cells, and microfluidic channels. Three fluorophores with different spectral characteristics are utilized to perform multicolor microscopy. According to the spectrum analysis of the fluorophores, a multicolor widefield fluorescence microscope is developed. Spectral crosstalk is corrected by exciting the fluorophores in a round-robin manner and synchronous emitted light collection. To report the performance of the multicolor microscopy, a simplified 3D tumor model is created by placing beads and spheroids inside a channel filled with the cell culture medium is imaged at varying exposure times. As a representative case and a method for bio-hybrid drug carrier fabrication, a spheroid surface is coated with beads in a channel utilizing electrostatic forces under the guidance of multicolor microscopy. Our experiments show that multicolor fluorescence microscopy enables crosstalk-free and spectrally-different individual image acquisition of beads, spheroids, and channels with the minimum exposure time of 5.5 ms. The imaging technique has the potential to monitor drug carrier transportation to cancer cells in real-time.

## Introduction

The functionalization of tetherless manipulatable micro-agents as drug carriers offers a remedy for required dose delivery to cancer cells. Microfluidic channels have been used as a controlled experimental environment to validate the effectiveness of such micro-agents [1, 2]. Since the channels are fabricated using optically transparent materials, microscopy is the most widely used method for imaging [3]. Among the microscopy techniques, fluorescence microscopy is the most effective way to visualize a specific target structure with a relatively high contrast-to-noise ratio by eliminating the background. In the literature, only micro-agents are

**Funding:** This work was supported by the European Research Council under the European Union's Horizon 2020 Research and Innovation programme under Grant 638428–project ROBOTAR: Robot-Assisted Flexible Needle Steering for Targeted Delivery of Magnetic Agents – European Research Council (ERC) Starting Grant.

**Competing interests:** The authors have declared that no competing interests exist.

imaged and localized using single-band fluorescence microscopy [4–10]. To study the drug transportation mechanism using micro-agents, imaging and localization of cancer cells and channels are also required. In this study, a multicolor widefield fluorescence microscope is developed to serially acquire images of micro-agents, cancer cells, and microfluidic channel. The main contribution of this study is that multicolor fluorescence microscopy is applied for spectrally-different and high-contrast visualization to monitor the interaction between micro-agents and cancer cells in a channel.

In numerous application domains ranging from single-cell visualization to rapid diagnosis, multicolor fluorescence microscopy is recognized as a key emerging technology [11–23]. The working principle of this microscopy technique is based on performing imaging using spectrally-different conventional fluorescent dyes [11], endogenous fluorophores [22], fluorescent proteins [24], or quantum dots [25]. Relatively narrow excitation and emission wavelength ranges enable image acquisition from different spectrum bands with multiple fluorophores. Acquired spectrally-different images, defined as individual fluorescence images, are overlaid to form a multicolor fluorescence image. Different configurations are developed to perform multicolor fluorescence microscopy. The simplest way is to collect multicolor images using a combination of an imaging sensor and single-band filters. Individual fluorescence images are collected by changing the filters with different bandwidths [23, 26, 27]. However, performing imaging by positioning the filters either manually or using a motorized drive creates a delay and limits the image acquisition rate. An appealing solution for stationary components is to use a multi-band emission filter [11, 15]. Since the delay associated with the spectrum band change is removed, the acquisition is limited by the frame rate of the imaging sensor. Depending on the spectrum of the fluorophores, the acquired images suffer from crosstalk [14]. To minimize the crosstalk, emitted light from the fluorophores is separated using dichroic mirrors and direct to individual imaging sensors using a common optical path [16, 18, 20–22]. Under simultaneous excitation of the fluorophores, the individual images still contain crosstalk due to the overlapping of emission spectra [17]. To overcome crosstalk, our multicolor microscope is configured to excite one fluorophore at a time and synchronously trigger the designated camera for individual image acquisition. We experimentally validate that the crosstalk is corrected by the assignment of emitted light from each fluorophore to the specified camera.

For imaging experiments, cancer cell spheroids with a diameter of 200 $\mu$m are formed using HeLa cells and placed inside a microfluidic channel filled with cell culture medium to create a simplified 3D tumor model. Polystyrene beads with a diameter of 20 $\mu$m are employed as surrogates for drug carriers and injected in the channel [28–30]. In our simplified tumor model, there are height differences between beads, spheroids, and channel. To form focused multicolor fluorescence images using widefield microscopy, individual image acquisition from different focal planes is required. Since micro-structures in the previously used samples have almost the same height, defocus aberration correction is not addressed for widefield multicolor imaging [11, 18, 31, 32]. In this study, our multicolor fluorescence microscope is specially developed for defocus aberration correction by spectrally-different image acquisition from multiple focal planes within the simplified tumor model. Fluorescein, CellTracker Red CMTPX, indocyanine green, fluorophores with different spectral properties, are used to render beads, spheroid, and channel visible by fluorescence imaging, respectively. Spectrum analysis of the fluorophores is carried out using a spectrofluorometer to determine effective wavelength ranges for excitation and fluorescence photon emission. Fluorophores are excited in a round-robin manner, and acquired individual images in one-round are overlaid to form time-lapse multicolor images. The achievable maximum multicolor image acquisition rate using the round-robin method is measured by exposure time analysis. Photobleaching curves of the fluorophores are studied to determine the duration of the multicolor imaging. To demonstrate

the application of the microscope for the investigation of drug targeting, HeLa cell spheroids are coated with micro-beads utilizing electrostatic forces. An optical flow method is used to compute motion information between consecutive time-lapse images. Our experiments show that multicolor fluorescence microscopy enables more information collection for localization and providing feedback by image acquisition from multiple spectrum bands compared to bright-field and single-band fluorescence microscopy.

## Materials and methods

### Spectrum analysis

Excitation and emission spectrum of the fluorophores are measured using a spectrofluorometer (FP-8300, Jasco, Japan) in the range of 200 nm and 900 nm with 1 nm data interval. For spectrum analysis, 2.5 $\mu$M indocyanine green (I2633–25MG, Sigma-Aldrich, USA) solution is prepared by dissolving 1.5 $\mu$g indocyanine green in 700 $\mu$L fetal bovine serum (16000044, Thermo Fisher Scientific, USA). The solution is placed in a water bath at 37˚C for 2 hours to bind indocyanine green to proteins in the culture medium [33]. A 10 $\mu$M working CellTracker Red CMTPX (Thermo Fisher Scientific, USA) solution is prepared in a serum-free medium by mixing 50 $\mu$g CMTPX dye in the vial with 7.29 $\mu$l dimethyl sulfoxide (D2650–100ML, Sigma-Aldrich, USA). Micro-beads stained with fluorescein (42–00-204, Micromod, Germany) are obtained in a solution form. 2 $\mu$L CMTPX and micro-beads solutions are diluted with 700 $\mu$L Milli-Q water and placed in quartz cuvettes (CV10Q700F, Thorlabs, USA) for the analysis. The spectrum graphs are illustrated in Fig 1(a). The peak excitation and emission wavelengths are measured as 492 nm and 513 nm for fluorescein, 583 nm and 611 nm for CMTPX, and 773 nm and 799 nm for indocyanine green, respectively. The multicolor widefield fluorescence microscope is developed based on the spectrum analysis for effective excitation of the fluorophores and accurate emitted light collection.

### Multicolor fluorescence microscope architecture

The microscope consists of illumination and emission units. The illumination unit generates discrete light beams with the center wavelengths of 470 nm, 565 nm, and 780 nm for sharp excitation of the fluorophores using three individual narrow-spectrum light-emitting diodes (LEDs) (M470L3, M565L3, M780LP1, Thorlabs, USA). The spectrum of LEDs is plotted in Fig 1(f). LEDs are collimated using 20 mm focal length aspheric condenser lenses (ACL2520U-A, ACL2520U-B, Thorlabs, USA) and coupled with filters (ET470/40x, ET572/35x, ET775/50x, Chroma, USA) for selection of the excitation wavelengths. Generated three discrete and collimated light beams are combined using two dichroic mirrors (T660lpxrxt, T510lpxrxt, Chroma, USA) for excitation of the fluorophores using a single optical path. The spectrum for the excitation path is shown in Fig 1(c). Collimated light provides non-homogeneous illumination, which results in artifact formation in the acquired fluorescence images. To achieve uniform illumination, Köhler illumination is used [34]. Collimated light is focussed for Köhler illumination using a 150 mm focal length plano-convex lens (LA1417, Thorlabs, USA). Köhler optics is constructed with the combination of a 150 mm focal length bi-convex lens (LB1374, Thorlabs, USA), a 75 mm focal length plano-convex lens (LA1145, Thorlabs, USA), and two iris diaphragms (SM2D25D and CP20S, Thorlabs, USA). The incoming light is first focussed and then collimated using bi-convex and plano-convex lenses, respectively. Images of the LED chips are obtained on the front focal plane of the plano-convex lens. A protective silver mirror (PF10–03-P01, Thorlabs, USA) is utilized to direct the collimated light. A 10x long working distance objective lens (Plan Apo, Mitutoyo, Japan) is employed to focus the excitation light on the sample. The objective is placed in a way that its back focal plane intersects with the

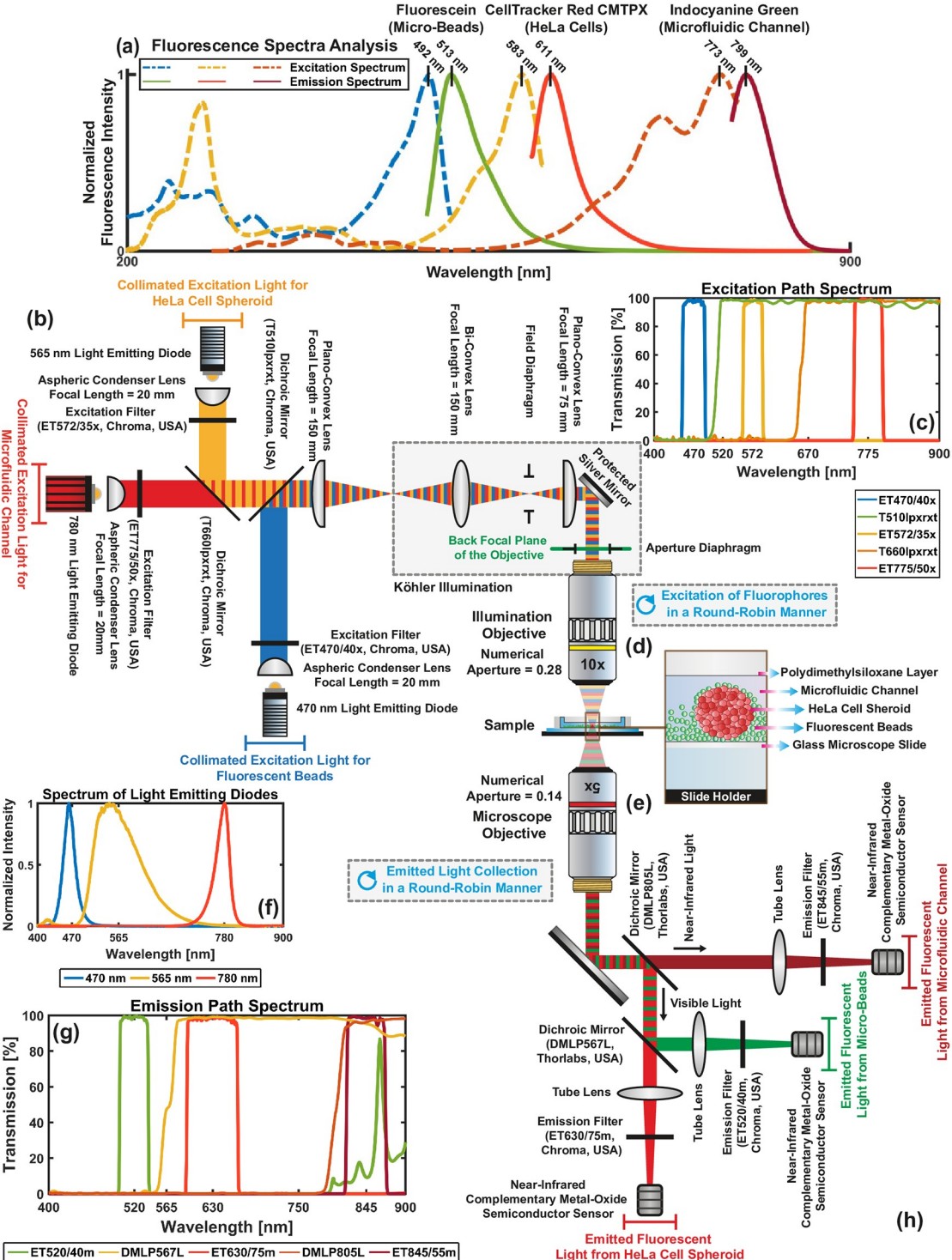

**Fig 1. Optical layout of the multicolor fluorescence microscope developed for monitoring interaction between polystyrene micro-beads as surrogates for drug carriers and cancer cells in a microfluidic channel.** (a) Two-dimensional excitation and emission spectrum of fluorescein, CellTracker Red CMTPX, and indocyanine green measured by a spectrofluorometer in the range of 200 nm and 900 nm. The peak excitation and emission wavelengths are shown with vertical lines. (b) and (h) Illumination and emission units. (c) and (g) Spectrum of excitation and emission paths, respectively. (d) and (e) Illumination and microscope objectives that employed for focusing excitation light on the sample and collection of emitted light from the fluorophores, respectively. (f) Spectrum of light-emitting diodes.

image plane of LED chips. To control illumination area and intensity at the sample plane, two individual irises (i.e., field and aperture diaphragms) are placed to back focal planes of plano-convex and objective lenses, respectively. Emitted light from fluorophores is collected using a 5x long working distance objective (Plan Apo, Mitutoyo, Japan) and transmitted towards the emission unit by a protective silver mirror (PF20–03-P01, Thorlabs, USA). The emission unit separates the emitted fluorescence light into its spectral components for individual image formation of microfluidic channels, HeLa cells, and micro-beads using two dichroic mirrors (DMLP567L and DMLP805L, Thorlabs, USA). Individual fluorescence light beams are directed onto tube lenses (ITL200, Thorlabs, USA) to form images onto complementary metal-oxide-semiconductor (CMOS) cameras (CS135MUN, DCC3240N, Thorlabs, USA). To block excitation light and select fluorescence wavelengths, emission filters (ET845/55m, ET623/60m, ET520/40m, Chroma, USA) are placed in front of the cameras. The spectrum for the emission path is plotted in Fig 1(g).

## Demultiplexer interface

Simultaneous excitation of the fluorophores leads to crosstalk between acquired fluorescence signals due to spectral overlap (Fig 1). To prevent crosstalk, the microscope is coupled with a demultiplexer interface to excite fluorophores in a round-robin manner and synchronically acquire fluorescence images using two individual pulse trains. Multicolor fluorescence images are obtained by overlapping serially acquired images from three different spectrum ranges in one round. The demultiplexer interface architecture and timing diagram are illustrated in S1 Fig. Pulse train signals are generated using a programmable dual-channel signal generator (33510B, Keysight, USA). A demultiplexer integrated circuit (74HC4052, Texas Instruments, USA) is used to distribute generated pulses to the corresponding LED driver (LEDD1B, Thorlabs, USA) and CMOS camera by channel selection using a microcontroller. Line drivers and voltage followers are connected to the inputs and outputs of the circuit to match the impedance and provide isolation, respectively.

## Cell culture and spheroid formation

HeLa cells are cultured in Dulbecco's modified Eagle's medium (11–965-092, Fisher Scientific Ltd. Canada), supplemented with 10%, fetal bovine serum (F7524–500ML, Sigma-Aldrich, USA), and 1% penicillin-streptomycin (15–140-122, Thermo Fisher Scientific Inc., USA). During the culture period, the cells are maintained at 37˚C in a humidified atmosphere containing 5% carbon dioxide. In the 2D cell culture, the medium is changed every 48 hours. When cells reach 80% confluency, the cells are trypsinized, counted, and resuspended in cell culture medium at a concentration of $2 \times 10^6$ cells/ml. In the next step 0.2 mL of cell suspension is then placed on top of an agarose mold, which was previously placed in the well of a 12-well plate for spheroid formation (Fig 2(a)). This results in spheroids which on average contain approximately 270 cells. Half of the medium is replaced daily. For measuring the spheroid morphology, images are taken every day (Fig 2(b)). The measured spheroid perimeter is used to determine the average spheroid diameter by calculating the surface area.

## Agarose molds

Agarose microarrays for nonadherent 3D cell culture are formed by replica molding [35]. Elastomeric stamps of polydimethylsiloxane (PDMS) (Sylgard 184 Silicon Elastomer Kit, Dow Corning, USA) are molded over the SU-8 photoresist master. Before use, the stamps are sterilized by 30 minutes incubation in 70% ethanol (84010059.5000, Boom Lab, the Netherlands). Afterward, 3% agarose (weight/volume) (16500500, ultra-pure agarose, Invitrogen, USA) is

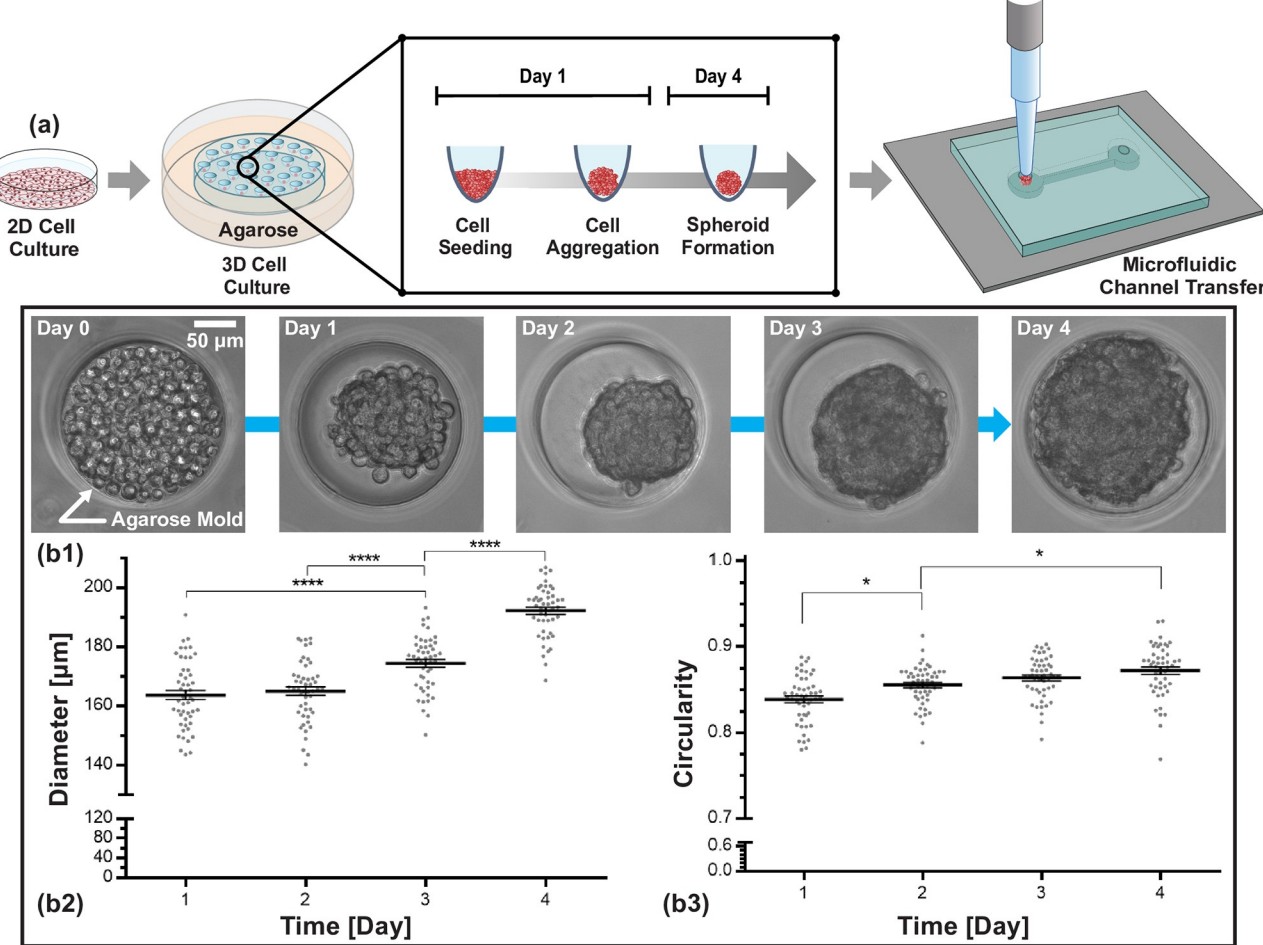

**Fig 2. HeLa cell spheroid formation.** (a) Schematic of cell spheroid formation in microwell array from micropatterned agarose well. (b) After the expansion of HeLa cells in standard tissue culture plates, cells were transferred into agarose microarrays for 3D cell spheroid formation with a diameter of 200 $\mu$m. Circularity and diameter of the spheroid versus time are plotted in (b2) and (b3), respectively.

cast onto the PDMS stamps and placed into 12-well round-bottom plates after polymerization. Each agarose mold contains 1500 microwells with depth and a diameter of 200 $\mu$m.

## Cell labelling using CMTPX and fixation

The 10 $\mu$M CMTPX working solution is prepared in a serum-free medium and subsequently incubated for 30 minutes with the cells in monolayer and spheroids which have compacted for 3 days under cultivation conditions. Afterward, the working solution is removed, and the culture medium is added again. For long-term storage, HeLa cell spheroids stained with CMTPX are fixated for 15 minutes with 4% formaldehyde (F8775–25ML, Sigma-Aldrich, USA) at room temperature. This is followed by washing twice with Dulbecco's phosphate-buffered saline.

## Microfluidic channel fabrication

To create a simplified 3D tumor model, HeLa cell spheroids stained with CMTPX are embedded in microfluidic channels with a height of 187 μm using a micropipette. The channels are fabricated using a standard soft lithography process [36]. Negative molds of microfluidic

channels on a silicon wafer are prepared in the cleanroom using SU-8 photoresist. A mixture of 10:1 Sylgard 184 PDMS and the curing agent is poured onto the wafer and cured in the oven at 70˚C overnight. The cured PDMS layer is gently removed from the wafer, and inlet-outlet ports are punched. Finally, the microfluidic channels are obtained by bonding the PDMS layer to a microscope slide using plasma-oxidation. For imaging experiments, the microfluidic channels are filled with 250 $\mu$g/ml indocyanine green in the culture medium [37].

### Image acquisition

The image acquisition and post-processing software is implemented in MATLAB (version R2017b, Mathworks Inc., USA) and run on a 64-bit Windows 10 computer, which has an Intel (R) Core(TM) i7–6700 CPU running at 3.40 GHz and 32 GB of RAM. The signal generator is configured through Ethernet communication to generate pulse trains for external trigger and exposure time. 8-bit grayscale fluorescence images with the size of 1280 × 1024 pixels captured by CMOS cameras are transmitted to the computer through the USB 3.0 interface. In our samples, micro-beads and microfluidic channels are located at different image planes with respect to HeLa cell spheroids due to height differences and lens aberrations. For multicolor fluorescence microscopy, the objective lens is focused on the image plane of the spheroids. Focused fluorescence images of beads and channel are obtained by placing each camera at a calibrated distance away from the tube lenses [38]. Misalignment between the acquired images is observed as a result of aberrations associated with pixel shift. For the alignment, a combination of kinematic mirror mount (KM200CP/M, Thorlabs, USA) and XY translation mounts (ST1XY-S/M, Thorlabs, USA), which allows pixel shift elimination by fine positioning, is coupled with CMOS cameras. An overview of our microscope assembly is shown in Fig 3.

## Results

### Multicolor fluorescence image acquisition

We report the performance of the multicolor fluorescence microscope by imaging micro-beads and HeLa cell spheroids placed inside a microfluidic channel. Fluorophores are excited between 450 nm and 490 nm, 554.5 nm and 589.5 nm, and 750 nm and 800 nm, for fluorescein, CMTPX, and indocyanine green, respectively (Fig 4(a)). To acquire 8-bit individual grayscale images of beads, spheroids, and channel with the minimum noise level, emitted light from the fluorophores is collected with 0 dB gain and 66 ms exposure time in the wavelength ranges of 500–540 nm, 592.5–667.5 nm, and 817.5–875.5 nm, respectively. Multicolor fluorescence images are formed by overlapping acquired individual fluorescence images with dimensions of 1231 $\mu$m × 983$\mu$m (horizontal × vertical). Fig 4 shows multicolor image formation for two representative cases. Acquired individual grayscale images are shown in Fig 4(b)–4(d) and formed multicolor images are shown in Fig 4(f). For visualization and interpretation of the multicolor images, intensity values in grayscale images of beads, spheroids, and channel are represented using black-green, black-red, black-blue color maps, respectively. To reveal further details in the multicolor images, the region of interests (ROIs) with dimensions of 521 $\mu$m × 417$\mu$m are magnified and shown in Fig 4(g). The channel contains higher beads concentration in Fig 4(g2) than Fig 4(g1). Although relatively high bead concentration leads to occlusion in the multicolor image (Fig 4(f2)), individual images include complete visualization of the beads, spheroids, and channel (Fig 4(b2)–4(d2)). Our experiments validate that multi-color microscopy enables occlusion-free visualization of the sample by spectrally-different image acquisition.

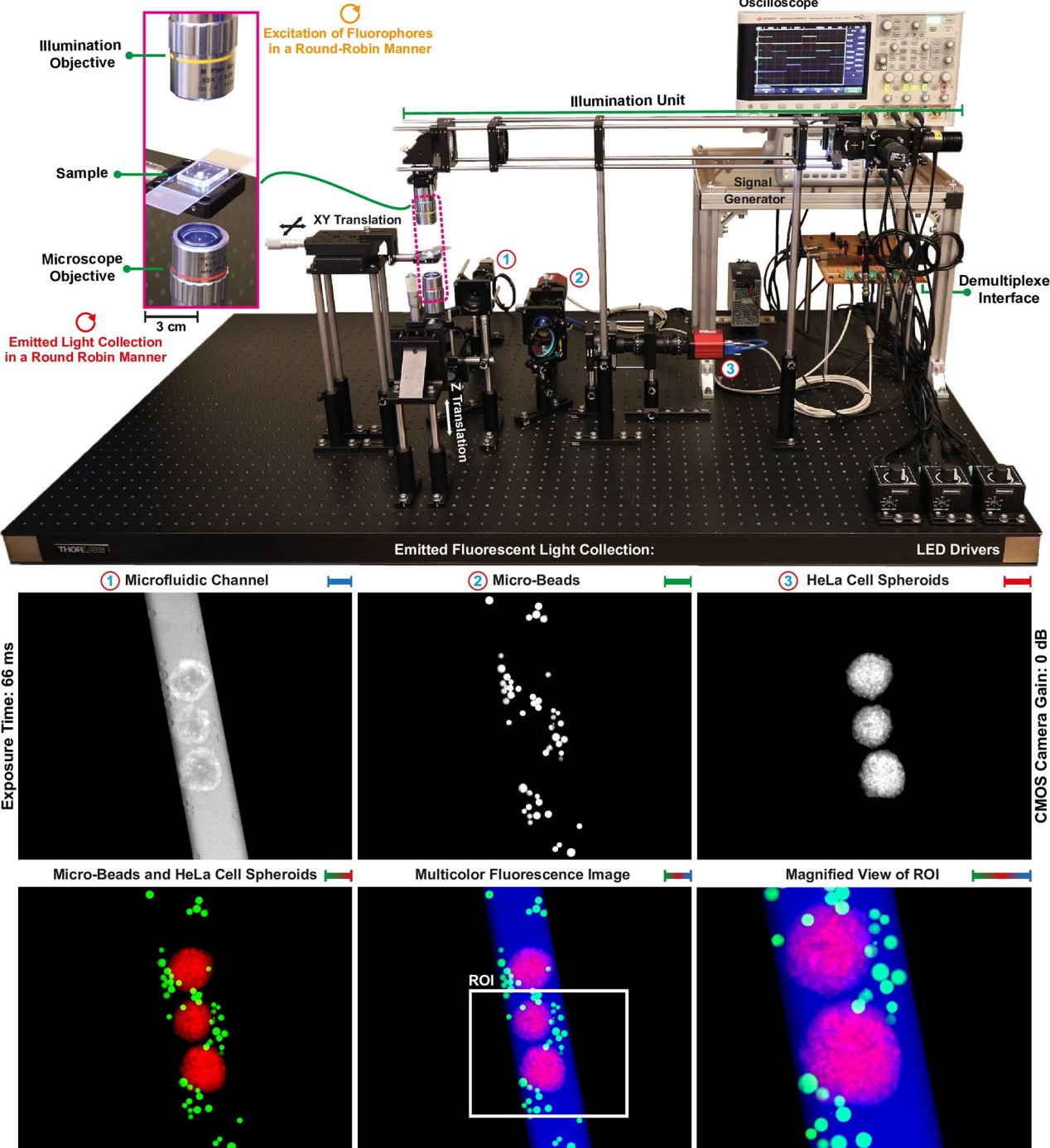

**Fig 3. Multicolor fluorescence microscope used for real-time image acquisition of microfluidic channel, HeLa cell spheroids, and micro-beads in a round-robin manner.** Scale bar: 100 $\mu$m.

## Correction of aberrations

We image our sample using commercial bright-field (Axio Vert. A1, ZEISS, Germany), fluorescence (EVOS M5000, Thermo Fisher Scientific, USA), and multicolor fluorescence microscopes to study correction of aberrations. For bright-field microscopy, the objective is focused

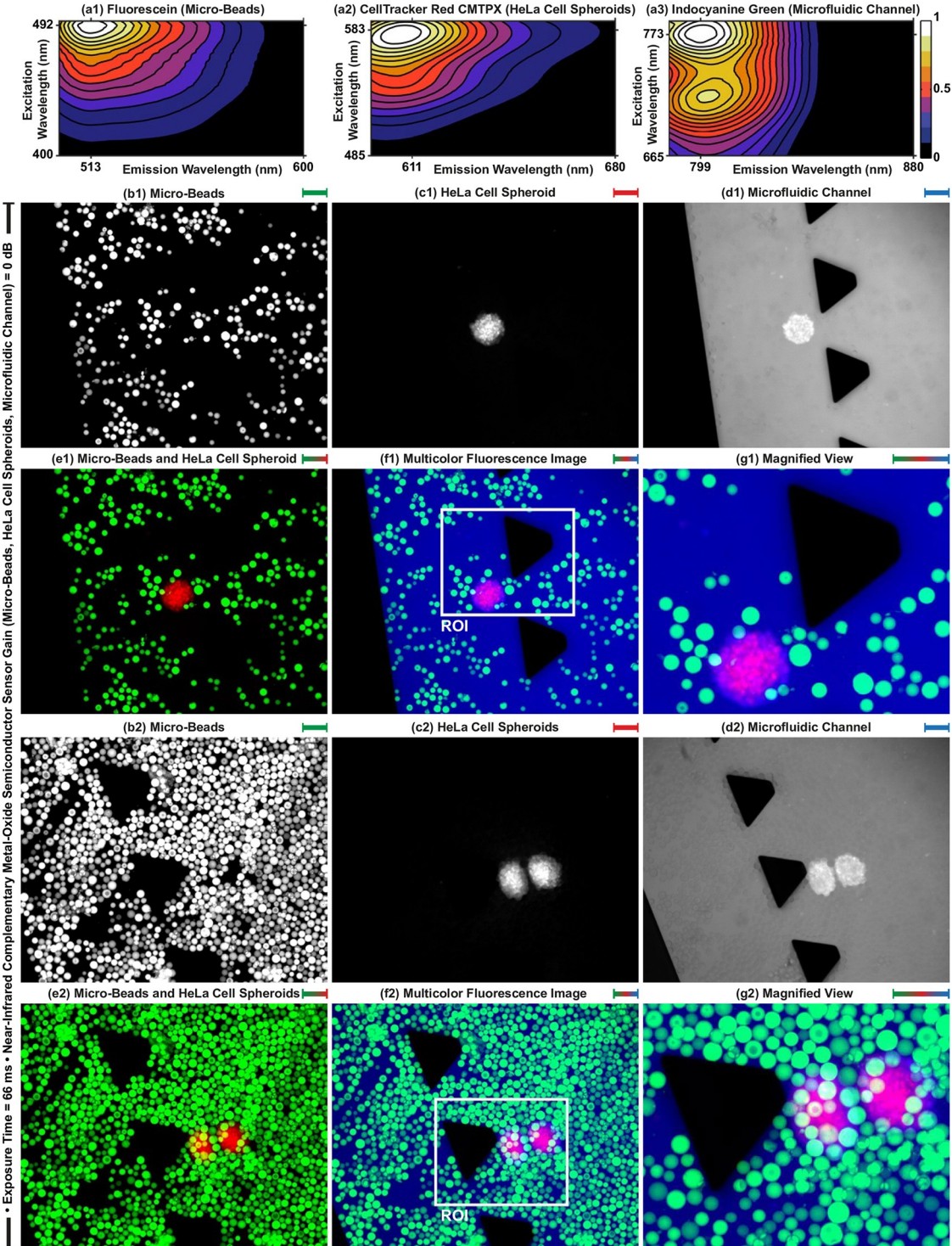

**Fig 4. Multicolor fluorescence image formation.** (a1)-(a3) Contour plots represent the spectrum of fluorescein, CellTracker Red CMTPX, and indocyanine green measured by a spectrofluorometer at varying excitation and emission wavelengths, respectively. (b)-(d) Acquired raw individual fluorescence images of micro-beads, HeLa cell spheroids, and microfluidic channel from different spectrum bands, respectively. (e) Merged fluorescence image of beads and spheroids. (f) Formed multicolor fluorescence image by overlapping acquired spectrally-different individual images. (g) Magnified view of the region of interest (ROI) defined on the multicolor image for revealing further details. Scale bar: 50 $\mu$m.

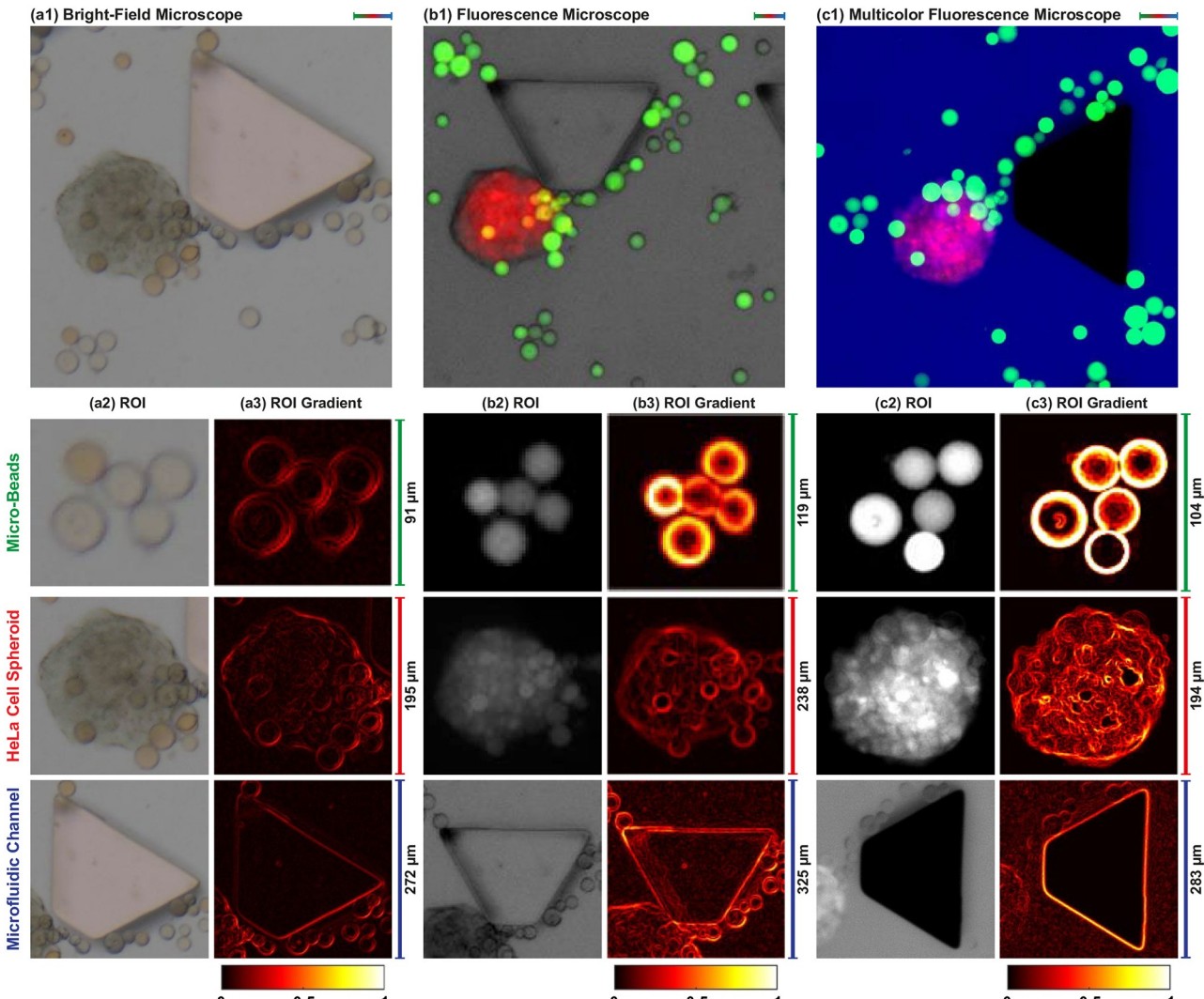

**Fig 5.** A microfluidic channel contains micro-beads and a HeLa cell spheroid is imaged with (a1) bright-field (Axio Vert. A1, ZEISS, Germany), (b1) fluorescence (EVOS M5000, Thermo Fisher, USA), and (c1) our multicolor fluorescence microscopes. (a2)-(c2) Magnified views of the region of interests (ROIs) cropped from the acquired microscope images. (a3)-(c3) Image gradients of the ROIs for contrast visualization. Scale bar: 50 $\mu$m.

on the spheroid. Image of the sample is captured using a color CMOS camera (BFLY-PGE-14S2C-CS, Point Grey, USA) (Fig 5(a1)). Height differences cause the image of beads to be blurred (Fig 5(a2)). Spectrally-different images are acquired with the fluorescence microscope by manually selecting color channels and individually focusing the objective on beads, spheroid, and channel (Fig 5(b1)). Since the microscope does not have optical components for indocyanine green, fluorescence images of micro-beads and HeLa cell spheroid are acquired. The microscope is also able to perform bright-field imaging. To obtain visualization of the channel, the bright-field image of the sample is acquired (Fig 5(b2)). Full visualization of the sample is obtained by overlapping acquired fluorescence and bright-field images. Multicolor fluorescence image of the sample is shown in Fig 5(c1). Our microscope produces focused image acquisition from different planes by adjusting the distance between the tube lens and CMOS cameras (Fig 5(c2)). Thus, the aberrations associated with the size difference between

micro-objects and lens aberrations are corrected for real-time multicolor fluorescence image acquisition.

## Crosstalk analysis for the multicolor fluorescence microscopy

Crosstalk analysis is carried out to visualize the effect of excitation light and emitted fluorescence signals on the acquired individual fluorescence images [14, 39]. Fluorophores are excited with single, double, and triple combinations of the center wavelengths at 470 nm, 565 nm, and 775 nm. Emitted fluorescence signals are simultaneously acquired from 500–540 nm, 592.5–667.5 nm, and 817.5–875.5 nm spectrum bands for micro-beads, HeLa cell spheroids, and microfluidic channel, respectively. According to the status of the center wavelength, we observe the following:

- 470 nm: ON, 565 nm: OFF, 780 nm: OFF (Fig 6(a1)): Fluorescence signal emitted from beads is collected without crosstalk from the 500–540 nm band. Relatively low-level crosstalk makes beads and spheroid visible in the image acquired from 592.5–667.5 nm band. No crosstalk is observed at the image acquired from the 817.5–875.5 nm band.

- 470 nm: OFF, 565 nm: ON, 780 nm: OFF (Fig 6(a2)): No crosstalk is observed at the images acquired from the 500–540 nm and 817.5–875.5 nm bands. Fluorescence signal emitted from spheroid is collected without crosstalk from the 592.5–667.5 nm band.

- 470 nm: OFF, 565 nm: OFF, 780 nm: ON (Fig 6(a3)) and 470 nm: OFF, 565 nm: ON, 780 nm: ON (Fig 6(a5)): No crosstalk is observed at the 500–540 nm band. Emitted light from the medium inside the channel is collected without crosstalk from the 817.5–875.5 nm band. The excitation light generated by 780 nm LED is collected from the 592.5–667.5 nm band and blocks the image acquisition.

- 470 nm: ON, 565 nm: ON, 780 nm: OFF (Fig 6(a4): Fluorescence image of beads is acquired from 500–540 nm band. The image acquired from the 592.5–667.5 nm band contain beads, spheroid, and channel as a result of crosstalk. Besides, excitation of CMTPX by 470 nm LED results in increasing the fluorescence signal intensity emitted from spheroid Fig 1(a). No crosstalk is observed at 817.5–875.5 nm band.

- 470 nm: ON, 565 nm: OFF, 780 nm: ON (Fig 6(a6)) and 470 nm: ON, 565 nm: ON, 780 nm: ON (Fig 6(a7)): Spectrum analysis shows that there is no overlap between fluorescein and indocyanine green (Figs 1(a) and 4(a)). However, the emission filter used for image acquisition of beads (ET520/40m, Chroma, USA) transmits the light with the wavelength range of 791–1000 nm (Fig 1(g)). Therefore, both beads and channel are visible at the image acquired from 500–540 nm. Excitation light generated by 780 nm LED blocks the image acquisition from 592.5–667.5 nm band. Fluorescence image of channel is acquired without crosstalk from 817.5–875.5 nm band.

The crosstalk is corrected and the blockage is removed by exciting the fluorophores in a round-robin manner and synchronically trigger cameras for individual image acquisition (Fig 6(a8)). The effect of crosstalk on the individual images acquired using continuous (Fig 6(a7)) and round-robin (Fig 6(a8)) methods is quantified by image quality assessment using structural similarity index measure (SSIM) metric [41]. Single-band fluorescence images of micro-beads (500–540 nm band), HeLa cell spheroid (592.5–667.5 nm band), and microfluidic channel (817.5–875.5 nm band) are used as references for the measurements (Fig 6(a1)–6(a3)). SSIM values for the individual images of beads (Fig 6(f1)) and spheroid (Fig 6(f2)) acquired

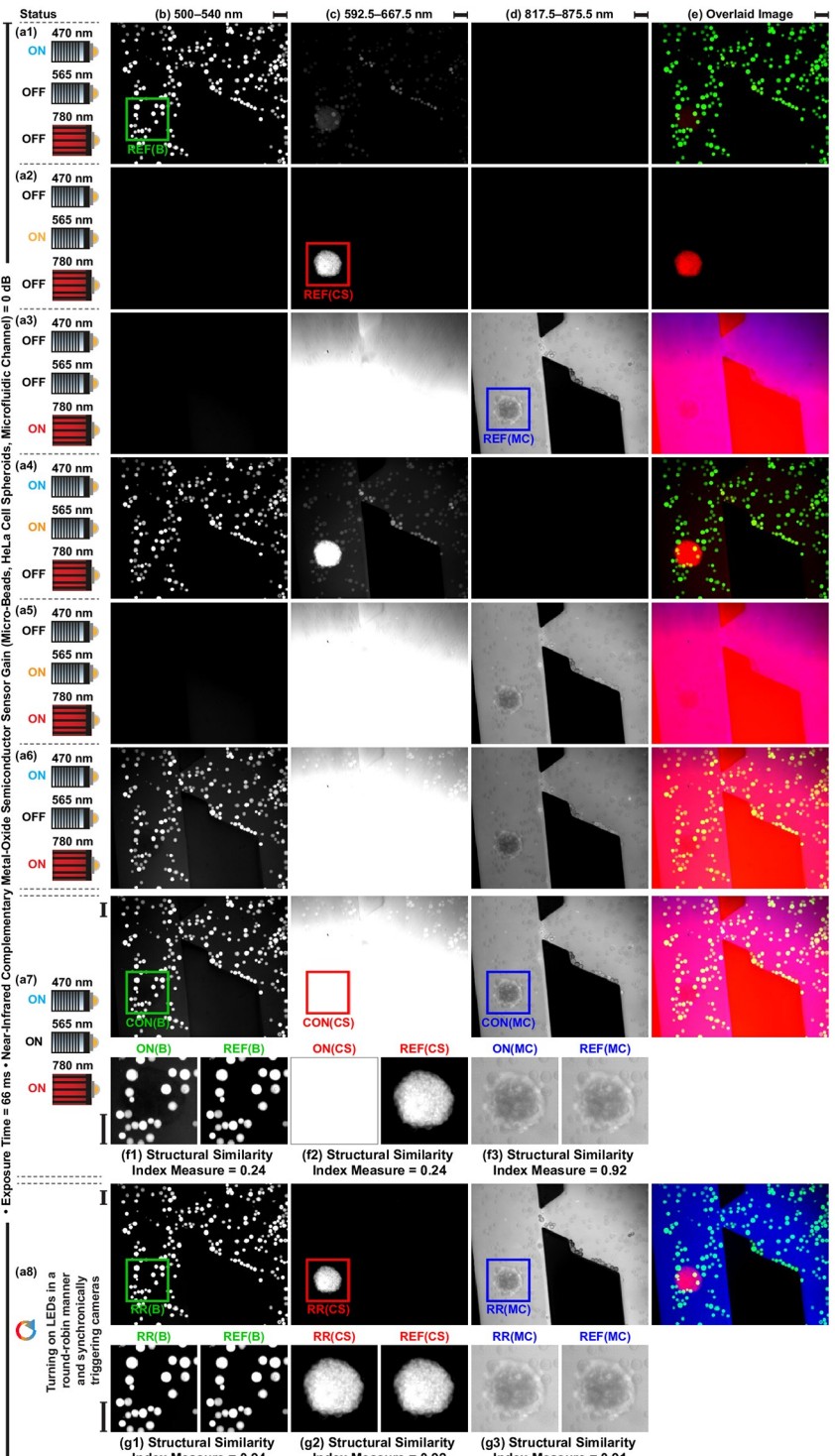

**Fig 6. Crosstalk analysis.** (a) Status of light-emitting diodes. (b)-(d) Acquired fluorescence images from 500–540 nm, 592.5–667.5 nm, and 817.5–875.5 nm spectrum bands, respectively. (e) Formed image by overlapping acquired images. (f) and (g) Quantitative crosstalk analysis by computing structural similarity measure index between reference images (REF) and images acquired using continuous (CON) and round-robin (RR) methods, respectively. Scale bar: 100 $\mu$m.

using the continuous method are computed as 0.24 due to the crosstalk and the blockage, respectively. Individual images of the channel are acquired using both continuous and round-robin methods with the SSIM value of about 0.91 (Fig 6f(3) and 6g(3)). Since indocyanine green is not able to be excited with 470 nm and 565 nm LEDs and the emission filter (ET845/ 55m, Chroma, USA) blocks the fluorescent photons emitted from fluorescein and CMTPX (Fig 1). We compute that the individual images are acquired using the round-robin method with the SSIM values of above 0.90, which enables spectral unmixing of micro-beads, HeLa cell spheroid, and microfluidic channel for dynamic real-time imaging (Fig 6(g)).

### Exposure time analysis

Exposure time ($\tau$), defined as the amount of time that a photodetector is subjected to the light for image formation, is the key factor for the maximum frame acquisition rate. For simultaneous multicolor imaging, the maximum frame acquisition rate equals $1/(\tau)$ [11, 13, 18]. Since a multicolor image is formed by the sequential acquisition of emitted fluorescence light from three different spectrum bands using the round-robin method, the maximum frame rate is computed as $1/(3\tau)$. To determine the maximum achievable frame rate, individual fluorescence images with the size of $1280 \times 1024$ pixels are captured with a 5 fps increment starting from 5 fps to 60 fps. The multicolor images at varying exposure times are shown in Fig 7(a). To reveal details of beads, spheroid, and pillar inside the channel, magnified views of the ROI defined on the multicolor image plane are shown in Fig 7(b). Both camera gain and LED power versus exposure time are plotted in Fig 7(c). Spectrally-different fluorescence images of micro-beads, HeLa cell spheroid, and microfluidic channel are acquired with 1 dB gains until 33 ms exposure time, which enables frame acquisition up to 10 fps without the noise associated with amplification. For enhancing fluorescence signal intensity, gains are increased to 5.3 dB when the exposure time is 33 ms for channel, 13.2 ms for beads, and 9.4 ms for spheroid, respectively.

To study the effect of exposure time on the multicolor fluorescence images, horizontal lines are defined on the beads, spheroid, and channel for contrast characterization [40]. Pixel intensities on the lines versus exposure time are plotted in Fig 7(d). When the exposure time is decreased from 66.0 ms to 5.5 ms for multicolor image acquisition with 60 fps, maximum pixel intensities of beads and channel drops by 13.3% and 25.6% at the full excitation power, respectively. No change is measured for the maximum pixel intensity of the spheroid since the increase in both LED power and camera gain bypasses the drop regarding exposure time. Peak signal-to-noise ratio (PSNR) and structural similarity index measure (SSIM) are used to quantify the multicolor image quality [41]. For both measurements, ROI acquired with 66.0 ms exposure time is used as the reference and computed values versus exposure time are plotted in Fig 7(e). We compute that decreasing exposure time from 33 ms to 5.5 ms reduces PSNR and SSIM by 27.6% and 12.9%, respectively. Our imaging experiments show that multicolor fluorescence images are able to be acquired at full sensor resolution up to 60 fps, which equals to maximum frame acquisition rate of the CMOS camera used for microfluidic channel imaging (DCC3240N, Thorlabs, USA).

### Uptake of indocyanine green by HeLa cell spheroids

Indocyanine green is a near-infrared fluorescent dye and one of its prominent application domains is cancer cells labeling [42, 43]. The microfluidic channels contain indocyanine green in the culture medium, which is a suitable environment for fluorescence labeling of cells. HeLa cell spheroids stained with CMTPX uptake indocyanine green after placed in the channels and becomes visible in the acquired individual fluorescence images of the microfluidic channel (Fig 4). Time-lapse imaging for dynamic uptake of indocyanine green in spheroids is shown in

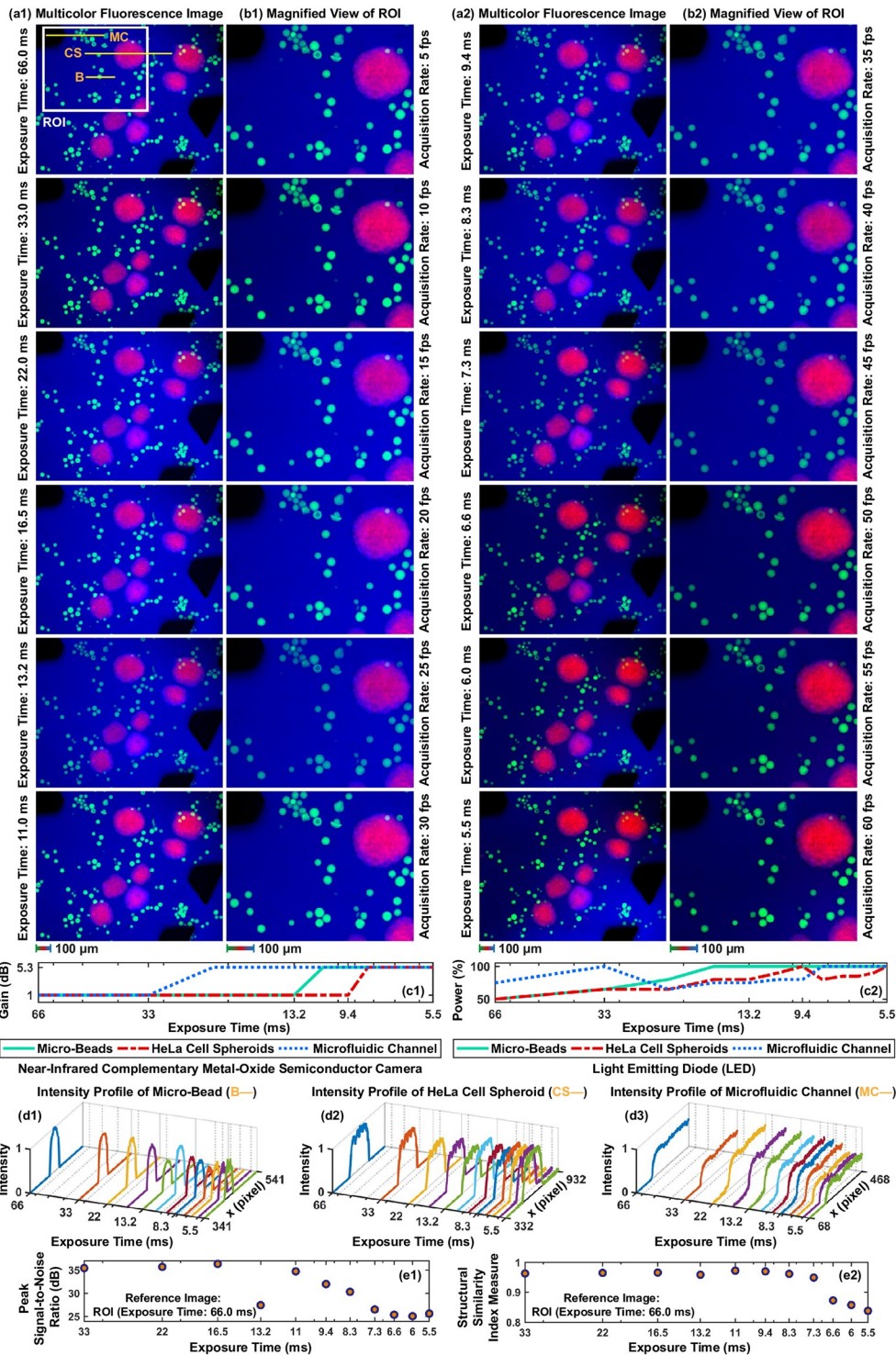

**Fig 7.** (a) Multicolor fluorescence images at varying exposure times. (b) Magnified view of the region of interest (ROI). (c1) and (c2) Camera gain and LED power during the image acquisition, respectively. (d1)-(d3) Intensity profiles of the lines defined on the beads, spheroid, and channel, respectively. (e1) and (e2) Measured peak signal-to-noise ratio and structural similarity measure index between multicolor images in (b) and the reference (multicolor image with 66.0 ms exposure time), respectively.

Fig 8(a) and 8(b). The effect of staining a spheroid with CMTPX and indocyanine green uptake on the formed multicolor fluorescence images are shown in Fig 8(c). CMTPX and indocyanine green enable dual-color fluorescence microscopy for the spheroid by image acquisition from 592.5–667.5 nm and 817.5–875.5 nm spectrum bands. Therefore, four-color fluorescence microscopy is performed utilizing fluorescein, CMTPX, and indocyanine green for dynamic imaging of micro-agents and 3D cancer cell spheroids in a microfluidic setup.

We observe that heating of the illuminated area results in a temperature gradient, which creates a flow inside the channel. To image the flow, a microfluidic channel contains 250 $\mu$g/ml indocyanine green in the culture medium is continuously excited. The acquired time-lapse image sequence is shown in Fig 8(d). For the first 81 minutes, indocyanine green bleaches so pixel intensities drop. The flow starts to bring the non-bleached fluorescent molecules area to the imaging area at 81-minutes, and pixel intensities start to increase (Fig 8(d2)). The effect of the flow on a spheroid is shown in Fig 8(b). Indocyanine green uptake is saturated at 30-minute and then the fluorescence signal decreases until 95-minutes as a result of photobleaching. The spheroid starts to uptake the non-bleached indocyanine green molecules brought by the flow at 95-minutes (Fig 8(b2)). Our imaging experiments show that the flow created by temperature gradient prolongs the imaging time of both microfluidic channels and HeLa cell spheroids at 817.5–875.5 nm band by recovering the fluorescence signals [44].

## Photobleaching resistance

When a fluorophore starts to be excited, emission light intensity decays over time. After a decay time, the emission of fluorescence photons is not possible due to irreversible photo-chemical alterations in the fluorophore molecules. This phenomenon is called photobleaching [45]. The photobleaching resistance of fluorescein, CMTPX, and indocyanine green is studied using time-lapse multicolor fluorescence microscopy of micro-beads, HeLa cell spheroids, and microfluidic channels, respectively. Multicolor fluorescence images are acquired under constant excitation power for 60 minutes with 66.0 ms, 33.0, 22.0, and 16.5 ms exposure times (Fig 9(a)). To express the effect of the photobleaching mechanism on the emitted fluorescence intensity, the normalized average intensity profiles of beads, spheroids, and channels over time are plotted in Fig 9(b). When the exposure time is 16.5 ms, we compute that the normalized average intensity of beads, spheroids, and channels decreases to 0.8 at 1.1 minutes, 29.5 minutes, and 60.0 minutes, respectively. Compared to fluorescein and indocyanine green, CMTPX exhibits a greater photobleaching resistance. Although beads and channels are not fully visible in the fluorescence images due to photobleaching, the spheroids are imaged with the normalized average intensity of more than 0.8 (Fig 9(a3) and 9(a4)). We also observe that the flow created by temperature gradient and indocyanine green uptake by spheroids create a fluctuation in the intensity profile of the microfluidic channel (Figs 8 and 9(b3)). The flow also fully recovers fluorescence imaging of microfluidic channel when the exposure time is 66.0 ms. Fluorescence recovery is not observed when the exposure time is lower than 66.0 ms. Imaging with higher exposure times increases the rate of recovery owing to a decrease in the photodamage on the fluorescence molecules brought by the flow [44].

## Photodamage analysis

Performing excitation in a round-robin manner inherently prolongs the fluorescence imaging time by reducing the photodamage on the fluorophores. To monitor the effect of photodamage on fluorescein, CMTPX, and indocyanine green, time-lapse fluorescence microscopy with continuous and round-robin excitation methods is performed for micro-beads, HeLa cell spheroids, and microfluidic channel, respectively (Fig 10). Photobleaching curves are obtained

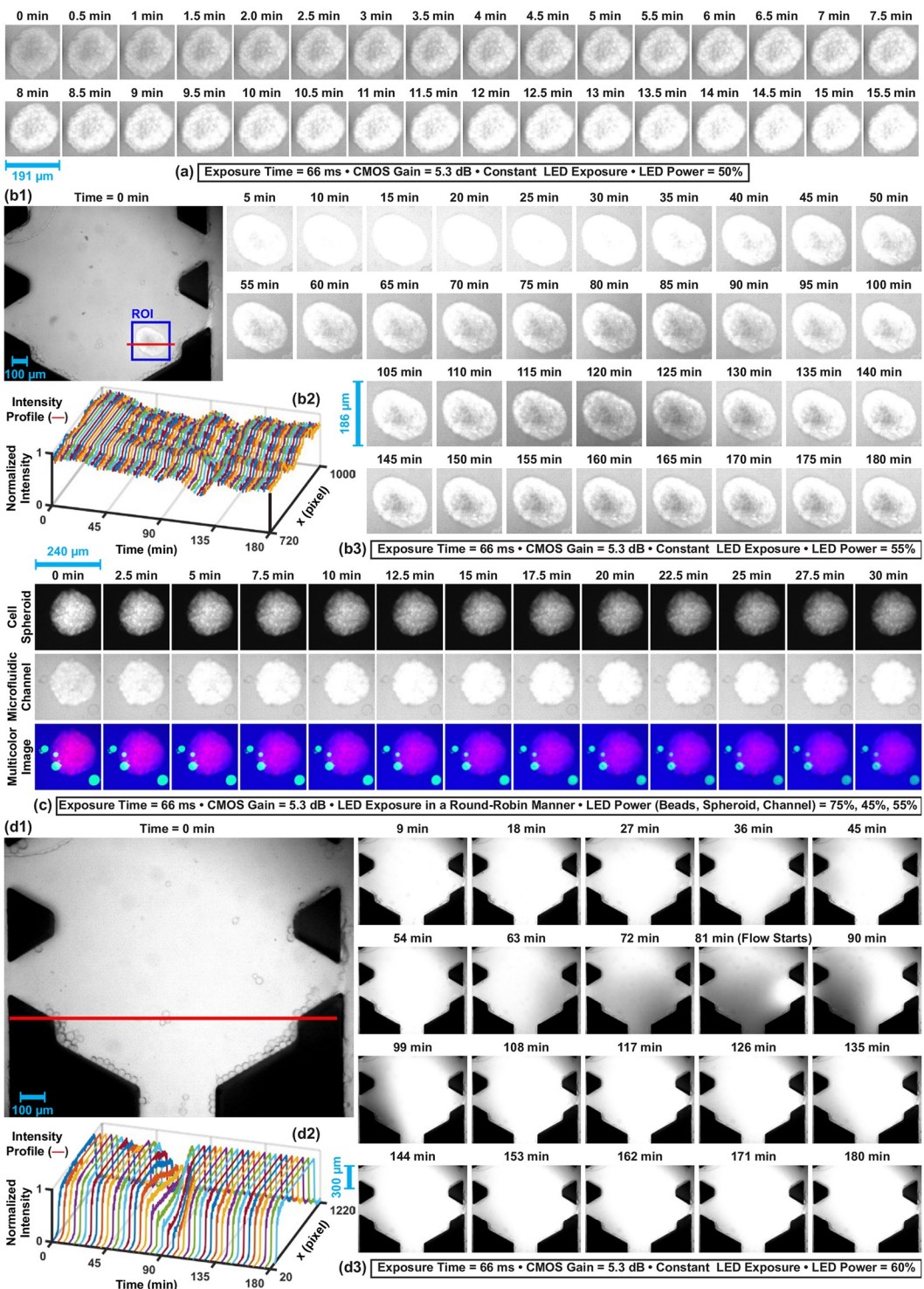

**Fig 8.** (a), (b) Time-lapse imaging for dynamic uptake of indocyanine green in a HeLa cell spheroid. (c) Effect of staining a spheroid using both CellTracker Red CMTPX and indocyanine green on the multicolor fluorescence images. (d) Time-lapse imaging of the flow created by temperature gradient. (b2) and (d2) Expressing effect of the flow on the fluorescence signal by plotting normalized pixel intensities over time on the defined red-lines in (b1) and (d1), respectively.

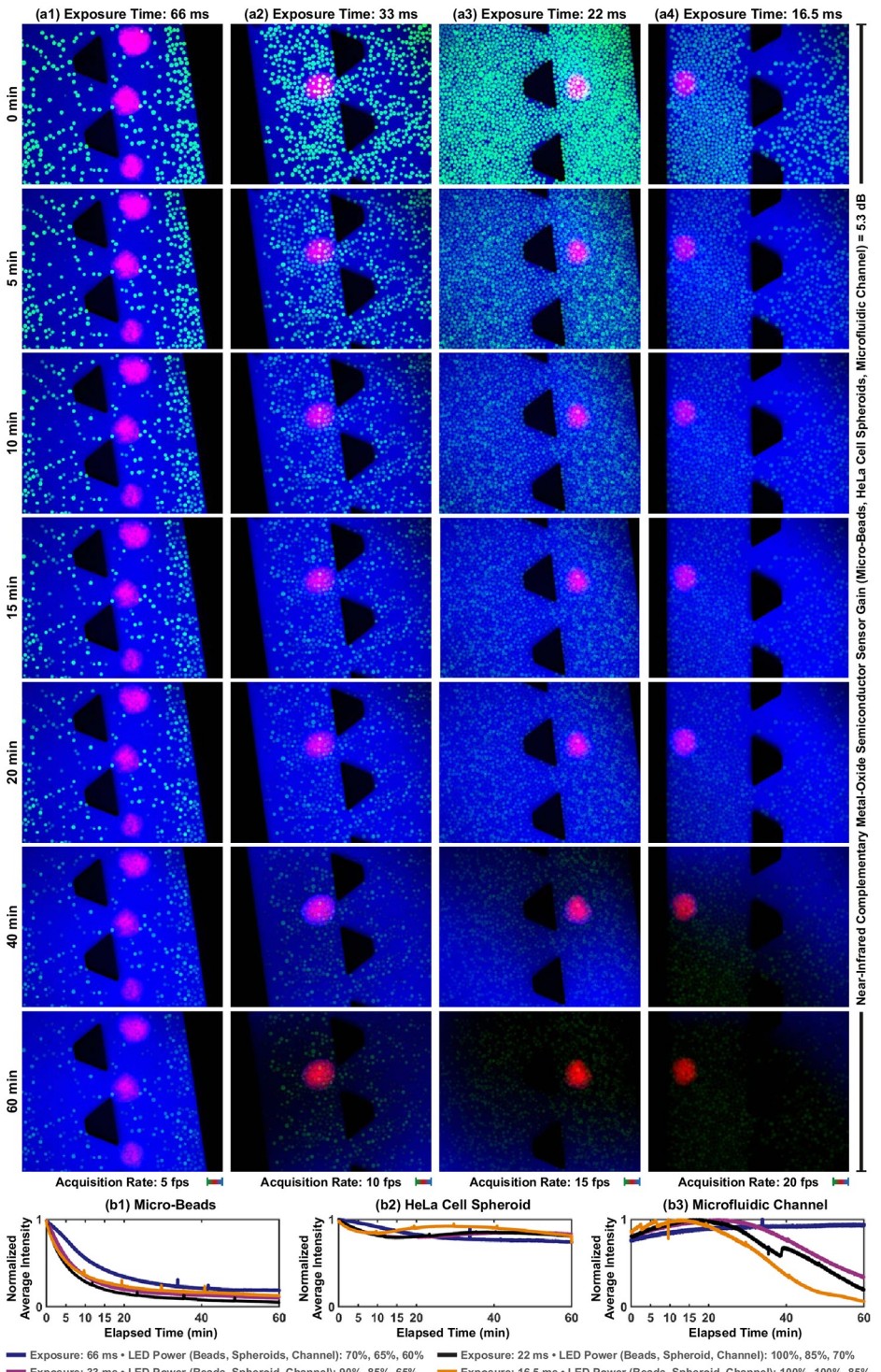

**Fig 9. The effect of photobleaching mechanism on multicolor fluorescence microscopy under constant excitation power.** (a) Time-lapse multicolor fluorescence images at varying exposure times. (b1)-(b3) Normalized average intensity profiles of micro-beads, HeLa cell spheroids, and microfluidic channel, respectively. Scale bar: 100 μm.

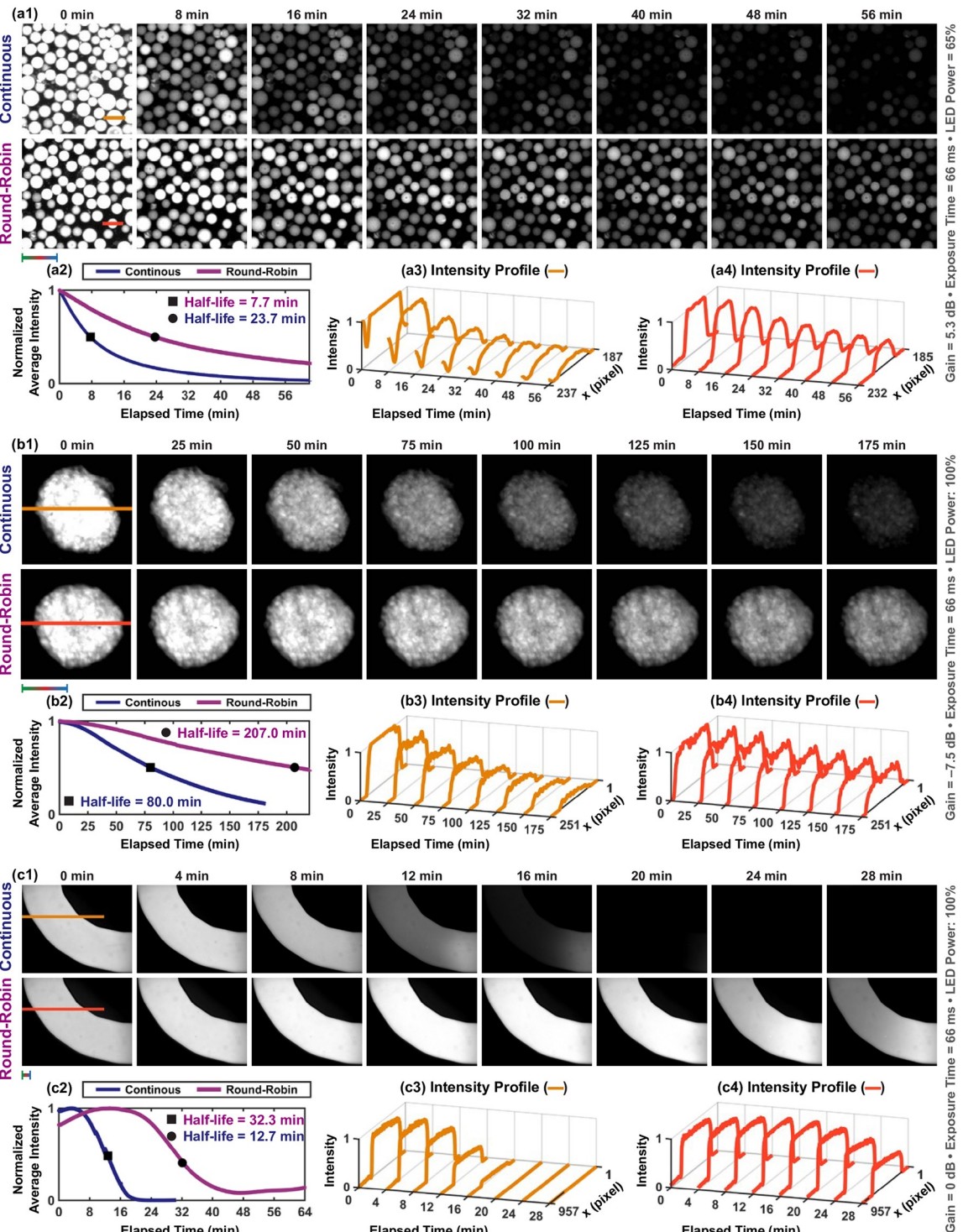

**Fig 10. The effect of continuous and round-robin excitation methods on photodamage mechanism.** (a1)-(c1) Acquired time-lapse fluorescence images of micro-beads, HeLa cell spheroids, and microfluidic channel, respectively. (a2)-(c2) Photobleaching curves by computing normalized average pixel intensity in each image. (a3)-(a4)—(c3)-(c4) Intensity profiles of the lines defined on beads, spheroids, and channel, respectively. Scale bar: 100 $\mu$m.

by computing normalized average intensity over time for photodamage characterization (Fig 10(a2)–10(c2)). Our experiments show that the round-robin excitation method increases the fluorescence half-lives of beads, spheroids, and channels 2.0-fold, 1.6-fold, and 1.5-fold, respectively. Approximately 0.5-fold drop in the half-lives of spheroid and channel is measured since CMTPX and indocyanine green are also excited by LEDs with the center wavelengths of 470 nm and 565 nm as a result of spectral overlap, respectively (Fig 6). We also observe that the flow created by temperature gradient does not occur with the continuous excitation due to relatively high excitation power [44]. On the other hand, round-robin excitation enables flow generation and fluorescence signal recovery of the channel by decreasing the photodamage on the indocyanine green.

## Real-time multicolor fluorescence microscopy

HeLa cell spheroids and polystyrene micro-beads are negatively and positively charged, respectively. Utilizing electrostatic forces, beads are able to be attached to the surface of spheroids without functionalization, which a self-assembly method used for bio-hybrid micro-agent fabrication [46]. As a representative application, the attachment of micro-beads to a HeLa cell spheroid in a microfluidic channel is imaged using multicolor fluorescence microscopy. For the imaging experiment, the channel is fixed on the sample holder attached to a translation stage (PT3/M, Thorlabs, USA). The holder is manually and randomly moved in horizontal and vertical directions until no further bead capture by the spheroid takes place due to the neutralization (Fig 11). Individual fluorescence images of the beads, spheroid, and channel are acquired with 22 ms exposure time and 5.3 dB gain at 15 fps and overlaid to form multicolor images. The motion of beads, spheroid, and channel between consecutive individual fluorescence images are analyzed using the Lukas-Kanade optical flow technique [47]. The analyzed motion at 63 seconds and 518 seconds is represented as three independent vector fields and overlaid onto the fluorescence images (Fig 11(c)). Our experiment shows that individual motion of micro-agents, 3D cancer cell spheroids, and microfluidic setup is able to be imaged in real-time by spectrally different image acquisition.

## Conclusions

This study presents multicolor fluorescence microscopy to visualize polymeric drug carriers, organic bodies, as well as their surroundings. Polystyrene micro-beads and HeLa cell spheroids are employed as an example of polymeric drug carriers and organic bodies, respectively. We develop a simplified 3D tumor model to validate spectrally-different image acquisition of beads, spheroids, and channel, using multicolor fluorescence microscopy. A multicolor wide-field fluorescence microscope is developed as a tool to acquire images from three different spectrum bands using a common optical path. To correct the spectral crosstalk, fluorophores with fairly well-separated emission spectra are excited in a round-robin manner and the emitted light is synchronously collected. In this study, we experimentally validate that multicolor fluorescence microscopy enables crosstalk-free visualization of beads, spheroids, and channel by individual image acquisition from different spectrum bands with a minimum exposure time of 5.5 ms. Our measurements show that multicolor fluorescence microscopy using round-robin method provide clear visualization by containing direct segmentation of beads, spheroids, and channel. We utilize multicolor fluorescence microscopy for real-time visualization of the interaction between drug carriers and cancer cells in a microfluidic channel.

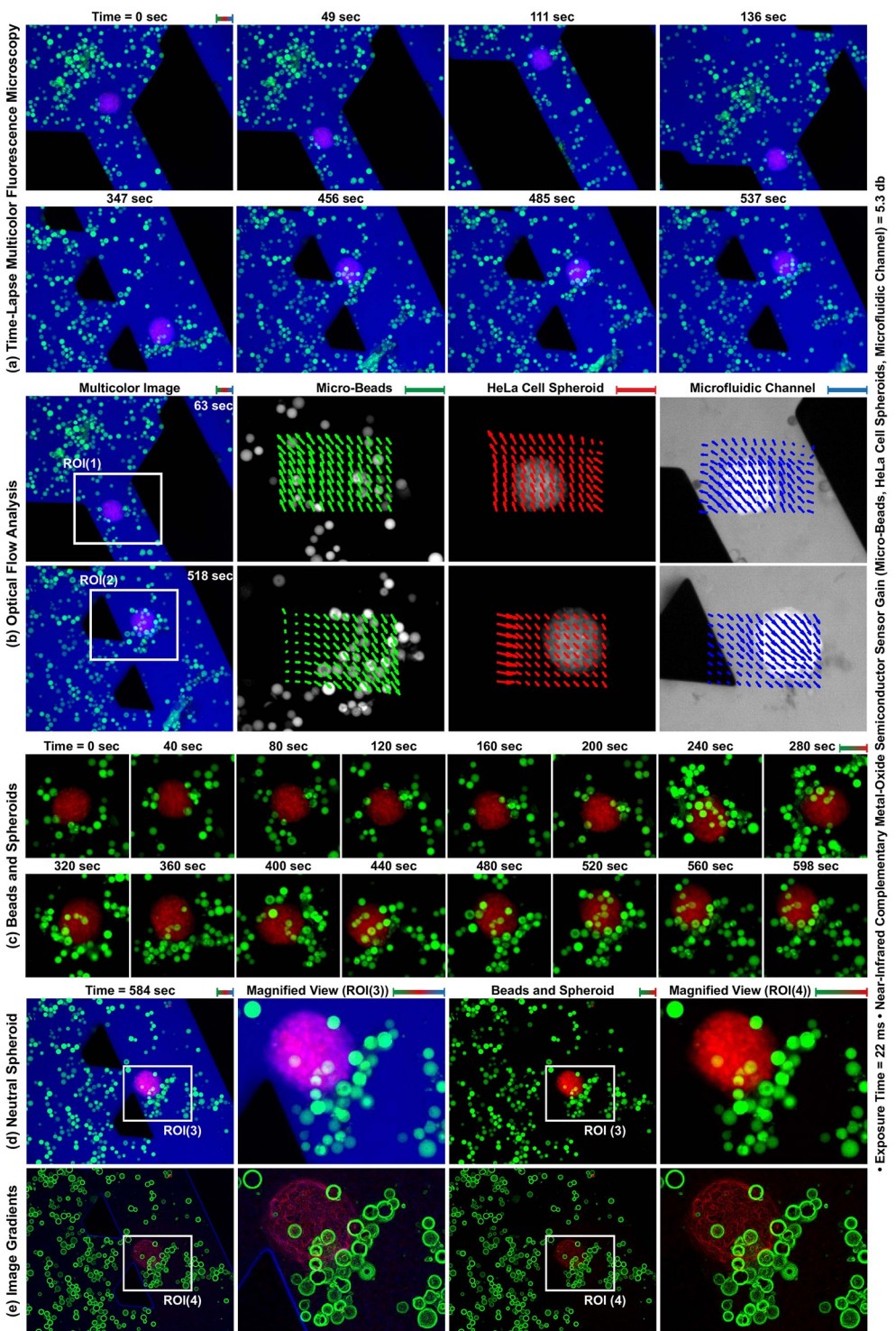

**Fig 11. Multicolor fluorescence microscopy for the attachment of polystyrene micro-beads to a HeLa cell spheroid surface utilizing electrostatic forces.** (a) Time-lapse multicolor fluorescence image sequence acquired at 15 frames per second. (b) The analyzed motion using Lukas-Kanade optical flow in ROI(1) and ROI(2) at 63 seconds and 518 seconds is represented as vector fields and overlaid on the fluorescence images. The spectrally different motion information of beads, spheroid, and channel is represented using green, red, and blue arrows, respectively. (c) Time-lapse fluorescence image sequence of beads and spheroid. (d) Neutral HeLa cell spheroid covered with micro-beads. (e) Gradients of the images in (d) for contrast visualization. Scale bar: 100 $\mu$m.

## Supporting information

**S1 Fig. Demultiplexer circuit diagram.** Schematic representation of the demultiplexer interface used for preventing crosstalk between fluorophores by synchronically triggering light-emitting diodes (LEDs) and complementary metal-oxide-semiconductor (CMOS) cameras in a round-robin manner using two individual pulse trains. (a and c) Architecture and computed frequency response plot of the demultiplexer circuit. (b) Timing diagram for input pulse trains generated using a signal generator (33510B, Keysight, USA) and output trigger signals for each LED and CMOS sensor combination.
(PDF)

## Author Contributions

**Conceptualization:** Mert Kaya, Fabian Stein, Jeroen Rouwkema, Islam S. M. Khalil, Sarthak Misra.

**Data curation:** Mert Kaya, Fabian Stein.

**Formal analysis:** Mert Kaya, Fabian Stein, Jeroen Rouwkema, Islam S. M. Khalil, Sarthak Misra.

**Funding acquisition:** Sarthak Misra.

**Investigation:** Mert Kaya, Fabian Stein.

**Methodology:** Mert Kaya, Fabian Stein, Islam S. M. Khalil.

**Project administration:** Mert Kaya, Fabian Stein, Sarthak Misra.

**Resources:** Mert Kaya, Fabian Stein.

**Software:** Mert Kaya.

**Supervision:** Jeroen Rouwkema, Islam S. M. Khalil, Sarthak Misra.

**Validation:** Mert Kaya, Fabian Stein, Jeroen Rouwkema, Islam S. M. Khalil, Sarthak Misra.

**Visualization:** Mert Kaya, Fabian Stein.

**Writing – original draft:** Mert Kaya, Fabian Stein.

**Writing – review & editing:** Mert Kaya, Fabian Stein, Jeroen Rouwkema, Islam S. M. Khalil, Sarthak Misra.

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
