## [Decision Letter · Decision Letter 0]

29 Apr 2021

PONE-D-21-10897

Serial imaging of micro-agents and cancer cell spheroids in a microfluidic channel using multicolor fluorescence microscopy

PLOS ONE

Dear Dr. Kaya,

Thank you for submitting your manuscript to PLOS ONE. After careful consideration, we feel that it has merit but does not fully meet PLOS ONE’s publication criteria as it currently stands. Therefore, we invite you to submit a revised version of the manuscript that addresses the points raised during the review process.

Keeping in mind that novelty is not included in PLOS ONE publication criteria, please review the feedback from reviewers and make any clarifying revisions.  

We look forward to receiving your revised manuscript.

Kind regards,

Kristen C. Maitland, Ph.D.

Academic Editor

PLOS ONE

Journal Requirements:

Reviewers' comments:

Reviewer's Responses to Questions

**Comments to the Author**

1. Is the manuscript technically sound, and do the data support the conclusions?

Reviewer #1: Yes

Reviewer #2: Partly

2. Has the statistical analysis been performed appropriately and rigorously? 

Reviewer #1: N/A

Reviewer #2: N/A

3. Have the authors made all data underlying the findings in their manuscript fully available?

Reviewer #1: Yes

Reviewer #2: No

4. Is the manuscript presented in an intelligible fashion and written in standard English?

Reviewer #1: Yes

Reviewer #2: Yes

5. Review Comments to the Author

Reviewer #1: In this manuscript the authors describe and evaluate a fluorescence microscope custom designed for efficient three-channel fluorescence imaging. The microscope design is thoroughly explained and presented by a diagram (Fig. 1). The custom multi-color microscope is demonstrated for imaging microparticles and cancer spheroids within channels and the results of characterization experiments for photobleaching, aberration artifacts, and spectral overlap are presented and discussed. The performance of the multi-color microscope is compared with a brightfield microscope and a traditional fluorescence microscope. Both qualitative and quantitative results are included.

Reviewer #2: Kaya et al presented a widefield microscopy scheme capable of imaging 3 fluorophores with distinct emission profiles. The developed system was demonstrated by imaging labeled tumor spheroid and fluorescent beads loaded in a microfluidic channel stained with a third fluorophore. My main concern is that the novelty of the manuscript is not convincing. The authors applied an existing microscopy technique to new samples, however no new biological findings are reported. Multicolor fluorescence imaging is very common and is extensively used in research labs.

Here are my other concerns/ suggestions:

1. Introduction, page 1, line 6-8. The authors state the advantages of using fluorescence microscopy. However the reasons stated can be true for any light microscopy technique and does not seem specific for fluorescence microscopy.

2. Introduction section, page 2, line 19-20, the authors mention that fluorescence microscopy is used for samples that are stained. This is not true as endogenous fluorophores or fluorescent proteins can imaged using this technique too. Also, rest of the paragraph consists cursory definition of fluorescence an its properties, which seems more appropriate for a tutorial style paper. Moreover, no references have been included.

3. HeLa cell spheroid with culture media does not “create a tumor micro-environment” which would involve other cells like immune cells, cancer associated fibroblast, extracellular matrix , nutrient, oxygen and pH gradients and so on. The authors develop a simplified 3D tumor model. Please consider re-phrasing

4. Please justify the diameters chosen for Hela cell spheroid and polystyrene beads. Are these clinically relevant values?

5. Why was the widefield microscope built with capability of imaging those 3 specific fluorophores? Development an adaptable system that could be used for multiple application would be more useful. Please justify.

6. Maybe I missed it but why was the microfluidic channel loaded with fixed HeLa cell spheroids instead of live cells, since fixed cells has no physiological relevance?

7. From figure 4(d1) it seems like there is bleed through of CellTracker Red CMTPX in the indocyanine green channel

8. Quantification of the data for cross-talk analysis would be more convincing along with the representative images shown.

9. The electrostatic interaction between beads and HeLa spheroids is confusing and does not provide additional data. The data shown are just images showing location of bead attachments on the spheroids. This does not necessarily provide information about the interaction.

10. Advantages over previously published multicolor widefield systems (eg: https://doi.org/10.1364/BOE.7.002285) should be discussed along with disadvantage or shortcomings of the presented technique. Also discuss the advantages of using a ‘round-robin’ detection scheme

6. PLOS authors have the option to publish the peer review history of their article (what does this mean?). If published, this will include your full peer review and any attached files.

Reviewer #1: No

Reviewer #2: No

---

## [Author Response · Author response to Decision Letter 0]

18 May 2021

Paper Title: Serial imaging of micro-agents and cancer cell spheroids in a microfluidic channel using multicolor fluorescence microscopy

Paper ID: PONE-D-21-10897

Dear Professor Maitland, 

In response to your email (dated 29 April 2021), we would like to thank you and the reviewers for the detailed comments. We have revised the letter accordingly, and the changes made are highlighted in red for clarity. We would like to thank the anonymous reviewers for their constructive comments.

Sincerely, 

Mert Kaya

Doctoral Candidate

Surgical Robotics Laboratory 

Department of Biomechanical Engineering

University of Twente, The Netherlands

E-mail: m.kaya@utwente.nl

Phone: +31-(0)53-489-1449

Website: www.surgicalroboticslab.nl

---

## [Editor Report · Decision Letter 1]

1 Jun 2021

Serial imaging of micro-agents and cancer cell spheroids in a microfluidic channel using multicolor fluorescence microscopy

PONE-D-21-10897R1

Dear Dr. Kaya,

We’re pleased to inform you that your manuscript has been judged scientifically suitable for publication and will be formally accepted for publication once it meets all outstanding technical requirements.

Kind regards,

Kristen C. Maitland, Ph.D.

Academic Editor

PLOS ONE
---

## [Editor Report · Acceptance letter]

4 Jun 2021

PONE-D-21-10897R1 

Serial imaging of micro-agents and cancer cell spheroids in a microfluidic channel using multicolor fluorescence microscopy 

Dear Dr. Kaya:

I'm pleased to inform you that your manuscript has been deemed suitable for publication in PLOS ONE. Congratulations! Your manuscript is now with our production department. 

Kind regards, 

on behalf of

Dr. Kristen C. Maitland 

Academic Editor

PLOS ONE